# Genome-Wide Association Study Revealed the Effect of rs312715211 in *ZNF652* Gene on Abdominal Fat Percentage of Chickens

**DOI:** 10.3390/biology11121849

**Published:** 2022-12-19

**Authors:** Yuting Zhu, Xiaojing Liu, Yongli Wang, Lu Liu, Yidong Wang, Guiping Zhao, Jie Wen, Huanxian Cui

**Affiliations:** 1State Key Laboratory of Animal Nutrition, Key Laboratory of Animal (Poultry) Genetics Breeding and Reproduction, Ministry of Agriculture, Institute of Animal Science, Chinese Academy of Agricultural Sciences, Beijing 100193, China; 2College of Animal Science and Technology, College of Veterinary Medicine of Zhejiang A&F University, Hangzhou 311300, China

**Keywords:** broilers, abdominal fat percentage, GWAS, SNP, *ZNF652*

## Abstract

**Simple Summary:**

With intensive selection in broilers, excessive abdominal fat accumulation is also present and causes economic concerns. Abdominal fat percentage (AFP) is one of the main indices of abdominal fat traits. We identified key SNP and candidate gene affecting AFP by a genome-wide association study (GWAS). Additionally, the main findings show that rs312715211 on the *ZNF652* gene can increase body weight (BW), reduce eviscerated carcass weight (ECW), and increase abdominal fat percentage (AFP).

**Abstract:**

Abdominal fat percentage (AFP) is an important economic trait in chickens. Intensive growth selection has led to the over-deposition of abdominal fat in chickens, but the genetic basis of AFP is not yet clear. Using 520 female individuals from selection and control lines of Jingxing yellow chicken, we investigated the genetic basis of AFP using a genome-wide association study (GWAS) and fixation indices (F*_ST_*). A 0.15 MB region associated with AFP was located on chromosome 27 and included nine significant single nucleotide polymorphisms (SNPs), which could account for 3.34–5.58% of the phenotypic variation. In addition, the π value, genotype frequency, and dual-luciferase results identified SNP rs312715211 in the intron region of *ZNF652* as the key variant. The wild genotype was associated with lower AFP and abdominal fat weight (AFW), but higher body weight (BW). Finally, annotated genes based on the top 1% SNPs were used to investigate the physiological function of *ZNF652*. Kyoto Encyclopedia of Genes and Genomes (KEGG) analysis suggested that *ZNF652* may reduce AFW and BW in broilers through the TGF-β1/SMad2/3 and MAPK/FoxO pathways via *EGFR* and *TGFB1*. Our findings elucidated the genetic basis of chicken AFP, rs312715211 on the *ZNF652* gene, which can affect BW and AFW and was the key variant associated with AFP. These data provide new insight into the genetic mechanism underlying AF deposition in chickens and could be beneficial in breeding chickens for AF.

## 1. Introduction

Carcass traits of chickens are economically important, and include abdominal fat (abdominal fat weight, AFW; abdominal fat percentage, AFP), body weight (BW), eviscerated carcass weight (ECW), etc. Broilers typically contain 150–200 g of fat per kg of BW. Abdominal fat accounts for 22% of body fat, but is generally considered waste, due to its low economic value [1]. Excessive deposition of abdominal fat in broilers not only results in low feed conversion rate, fertility, and semen quality, but also affects the economy of the industry [2,3,4]. Currently, decreasing abdominal fat deposition is one of the objectives of broiler production. AFP is one of the main phenotypic indices of abdominal fat traits [5] and is an important component of chicken breeding. However, the genetic basis of AFP remains unclear.

AFP has relatively high heritability, ranging from 0.53 to 0.71 [1,6,7]. This suggests that direct selection may be used in future breeding programs to reduce AFP in broilers. Studies have shown that the AFP of broilers can be reduced via genetic selection [8], and selected broilers show lower AFP and fat deposition than unselected commercial chickens do [9,10]. In a Northeast Agricultural University study, high and low AFP lines were selected, and after 23 generations of selective breeding, the AFP of broilers from the fat line was 9.87 times greater than that of broilers from the control line, despite the fact that there was no difference in body weight between the two lines [11]. These studies show that selection for AFP can reduce abdominal fat deposition in broilers while maintaining or increasing their body weight.

Genome-wide association studies (GWAS) have become the main approach to studying economically important traits in poultry [12]. Numerous statistically significant single nucleotide polymorphisms (SNPs) and qualitative trait loci (QTLs) have been found for AFP of broilers. For example, Hu et al. [13] found that two SNPS in the fat line were significantly associated with AFP after selection. There are reports that revealed a QTL region that significantly affected AFP and explained 6.24% of phenotypic variation [14]. However, to date, the major sites of genetic variation and candidate genes affecting AFP have not yet been revealed. We are, therefore, committed to revealing genomic regions, mapping QTLs, searching for loci and genes associated with abdominal fat deposition, and applying our findings to actual breeding production, in order to improve selective breeding in broilers.

A total of 520 female Jingxing yellow chickens (selection line: 258, control line: 262) were used in this study; among them, the selection line was selected for 16 generations according to intramuscular fat (IMF) content and body weight (BW) [15]. In previous studies, IMF, BW, and AFP were moderately phenotypically and genetically correlated [16]. The objectives of this study were to identify the major SNPs and candidate genes affecting AFP using GWAS and fixation index (F*_ST_*) analyses, in order to explore the genetic basis of AFP. We hope that effective variants can be identified and used in the practical breeding programs to improve the economics of broiler production.

## 2. Materials and Methods

### 2.1. Population and Sample Collection

A total of 520 female Jingxing yellow chickens (selected line: *n* = 252; control line: *n* = 268) were bred for intramuscular fat (IMF) and used in this experiment. After 16 generations of breeding, IMF and AFW in the selection line were significantly increased, and breeding programs were as described in Zhao et al. [15]. The chickens were immunized, reared, and maintained under uniform conditions and provided food and water ad libitum, in accordance with NRC international nutritional standards. Blood samples were collected from the chickens’ wings the day before slaughter, and EDTA anticoagulant was stored at −20 °C for DNA extraction. All experimental animals were slaughtered at 98 days of age, and phenotypic traits were measured from carcasses, including AFP, BW, ECW, half eviscerated weight, etc. Abdominal fat percentage was calculated as:AFP = AFW/(ECW + AFW) × 100%

### 2.2. Genotyping and Quality Control

DNA was extracted from the blood of 520 female Jingxing yellow chickens using the standard phenolic method, and DNA concentration was measured using nano-Drop1000 nucleic acid protein analyzer (Thermo Fisher Scientific Inc., Waltham, MA, USA). The determination standard was A260 nm/A280 nm (1.8–2.0), Qualified samples were sent to the Beijing Boao Jingdian Company for whole-genome resequencing with a depth of 10×. Double-terminal sequencing (PE150) was used based on the Illumina Novaseq 6000 sequencing technology platform. Whole genome resequencing data were filtered using FASTP (0.19.5) for quality control. The phred mass value was 30, each read had ≥75 bases, and the base mass was removed <30% base. The clean reads were compared with the chicken 6.0 reference genome using BWA software (http://bio-bwa.sourceforge.net, accessed on 10 July 2022). More details about the genotyping and filtering steps are available in the work of Liu et al. [17]. The accession data codes are CRA002643 and CRA002650 at https://bigd.big.ac.cn/gsa/ (accessed on 10 July 2022).

The original sequencing data were first filled with SNP loci using Beagle 5.0 software [18]. PLINK 1.9 was used for further quality control of phenotype and genotype data. After quality control (SNPs and individuals with detection rates >90%, minor allele frequency > 0.05, and extreme deviation from Hardy–Weinberg proportions (*p* > 0.00001)), 495 animals, 8,940,029 SNPs, and chromosomes 1–28 were retained.

### 2.3. Genome-Wide Association Study

We applied a linear mixed model (LMM) to conduct a GWAS examining the traits underlying AFP in the selection (*n* = 252) and control (*n* = 268) lines, using GEMMA software [19]. The population structure was assessed via principal component analysis (PCA), using PLINK 1.9 to correct for differences between the two lines. The genetic relationship matrix and the first three PCA were added for correction during the LMM calculation based on the model: y = Wα + xβ + u + e, where y represents the phenotypic values of n samples, W represents the fixed effect matrix, α represents the fix effect (population structure, including PCA1, PCA2, and PCA3), x represents the SNPs, β represents the effects of corresponding markers, u represents the random effect vector, and e represents a vector of random residuals. Finally, we analyzed single loci one-by-one and calculated *p*-values using a derived Wald test. The Bonferroni correction multiple test was used to determine the significance threshold after LD linkage modified SNPs. The sums of the independent LD blocks plus singleton markers were used to define the number of independent statistical comparisons [20]. Finally, 378,446 independent SNPs were used to determine the *p*-value thresholds, including genome-wide significance (−log10(0.05/378,446)) and suggestive association (−log10(1/378,446)). The Manhattan plots of GWAS for AFP were produced using the “qqman” package in R (https://www.r-project.org/) (accessed on 10 July 2022).

### 2.4. Fixation Indices (F_ST_) and Heterozygosity (π)

Selection signals combined with nucleotide polymorphism analysis can reveal the genetic mechanisms underlying population evolution. Based on the SNPs remaining after quality control, F*_ST_* and π analyses were performed on the selected and control lines using Vcftools (V0.1.13) [21]. In F*_ST_* analysis, a single SNP was used as the step size for genome-wide scanning. The top 5% of F*_ST_* values were defined as selected loci between the selection and the control lines. Then, π values were calculated for the significant SNPs screened by F*_ST_*. During π analysis, the window 40,000 and step 10,000 were used to calculate the region.

### 2.5. Dual-Luciferase Reporter Assay

To quantify the interaction of rs312715211 and its potential target gene *ZNF652*, we constructed vectors based on PGL4.18 plasmids. According to the chicken 6.0 reference genome sequence, we constructed a 1500 bp promoter sequence upstream of the transcription start site and designated it PGL4.18-ZNF652-pro. We also constructed wild and mutant luciferase reporter vectors of rs312715211 and designated them PGL4.18-SNP-TT and PGL4.18-SNP-AA. Next, chicken embryo fibroblasts (DF1) cells were cultured on a 24-well plate and were co-transfected with 250 ng of PGL4.18-ZNF652-pro and 250 ng of PGL4.18-SNP-TT or PGL4.18-SNP-AA. After 24 h, 100 μL lysate was added according to the Dual Luciferase Reporter Gene Assay Kit (Promega, Madison, WI, USA) and centrifuged for 15 min. Firefly and Renilla activities were measured using 20 ul supernatant, and each group was replicated three times.

### 2.6. RNA-Seq and Weighted Gene Correlation Network Analysis (WGCNA)

Transcriptomic sequencing was performed on 98-day-old Wenchang chickens’ abdominal fat, including 18 samples. After eliminating abnormal individuals, high and low AFP groups were selected. For detailed sequencing steps, please refer to previous studies [22]. Then, R package “WGCNA” was used to analyze the weighted gene correlation network of five transcriptomes from abdominal fat tissue of chickens with high or low AFP. A weighted co-expression network was constructed using β = 7 to calculate the adjacency between genes. In addition, parameters mergeCutHeight = 0.25 and minmodule-size = 30 were selected for calculation, and gene significance, correlation of modules, and gene expression profile were calculated. The RNA-seq data describing the abdominal fat are included in a previous report by our group and are available at https://bigd.big.ac.cn/gsa/ (accessed on 10 July 2022) (accession data code CRA006031).

### 2.7. Kyoto Encyclopedia of Genes and Genomes (KEGG)

The top 1% of SNPs were selected for gene annotation, and these genes were enriched. The genes of the significant modules analyzed from the WGCNA were also annotated and then enriched. The KEGG database (http://www.genome.jp/kegg) (accessed on 10 July 2022) [23] is an important public database used for metabolic analysis and regulatory network research. KEGG enrichment analysis of annotated genes was performed using KOBAS, and KEGG pathways with a Q value (corrected *p* value) ≤ 0.05 were considered significantly enriched. The results were drawn using the “ggplot2” package in R.

### 2.8. Statistical Analysis

R 4.0.4 and SAS 9.4 (SAS Institute, Cary, NC, USA) were used to generate descriptive statistics and for normal distribution tests of AFP and to test the significance of the differences between the groups using Student’s *t*-test. Confidence limits were set at 95%, and *p* < 0.05 (*) or < 0.01 (**) was considered significant. Data are presented as mean ± standard error. ASReml 3.0 software was used to estimate genetic variance, genetic correlation, and heritability. Genetic parameter estimation was based on the animal single trait model using restricted maximum likelihood (REML) and the model:

Y = Xb + Za + e, where Y represents the observed value of traits, b represents the fixed effect vector, and X represents the incidence matrices of fixed effects. a represents the additive genetic effect vector of the individual, Z represents the incidence matrices of the additive genetic effect of the individual, and e represents the random residual effect vector.

The phenotypic variation explained (PVE) can be estimated using equation [24]:$$\mathrm{PVE}=\frac{2\beta^{2}MAF\left(1-MAF\right)}{2\beta^{2}MAF\left(1-MAF\right)+(se\left(\beta \right)){\;}^{2}2NMAF\left(1-MAF\right)}$$
where *β* represents the effect value for the GWAS result, *MAF* represents SNP minor allele frequency, and N represents the number of individuals included in the GWAS analysis.

## 3. Results

### 3.1. Phenotypic Statistics and Heritability Evaluation

Descriptive statistics for AFP, ECW, and AFW for both the selection and control lines are presented in Table 1 and Figure 1. The results showed that the AFP, AFW, ECW, etc., of the selection line was higher than control line after selection. Genetic parameter analyses revealed that AFP has high heritability—the heritability was 0.64. Moreover, AFP, BW, and IMF showed positive genetic correlation and phenotypic correlation. The AFW and ECW were significantly (*p* < 0.01) increased in the selection line, compared to the control line, resulting in no significant difference in AFP (AFP correlates with the AFW/ECW ratio) between the selection and control lines.

### 3.2. GWAS Identified the Effective Variants and Candidate Genes

To find significant variation at the genomic level, we performed GWAS of AFP. The GWAS results are summarized in Figure 2 and Table 2. The chromosomal significance threshold was −log_10_(0.05/378,446). The potential significance level threshold was −log_10_(1/378,446). These analyses showed that the significant SNPs associated with AFP phenotype were located on chromosomes 1, 14, and 27. An approximately 0.15 MB region on chromosome 27 (Chr27: 5,963,734–6,119,680) was strongly associated with AFP and included nine significant SNPs with AFP, which were annotated on *IGF2BP1, ZNF652, GIP*, *UBE2Z*, and *ETV4*, respectively. The most significant SNP (rs312351828) accounted for 5.35% of the observed phenotypic variance. Taken together, all of the SNPs that reached the GWAS threshold explain 3.34−5.58% of the observed phenotypic variance.

### 3.3. rs312715211 in the Intron Region of ZNF652 Was the Primary Variant Associated with AFP

To further identify candidate SNPs and genes, the F*_ST_* and π values of all SNPs located on chromosome 1, 14, and the candidate region (Chr27: 5,963,734–6,119,680) were calculated. F*_ST_* is calculated with a single SNP as the step size, and the F*_ST_* threshold was 0.1. These data are summarized in Figure 3 and Figure 4. F*_ST_* analysis showed that SNPs reaching the GWAS threshold line on chromosomes 1, 4, and 14 were not selected (Figure 3).

On chromosome 27, F*_ST_* analysis showed that 159 SNPs in the target region were selected (Figure 4 and Appendix A), and this region included the *ZNF652, IGF2BP1, ETV4, DHX8, GIP, PHOSPHO1, SNF8, ABI3,* and *UBE2Z* genes. However, only rs312715211 reached the threshold for GWAS and F*_ST_* (Figure 4A). We calculated the π values in this region, in order to confirm whether the candidate sites were selected. The π values represent nucleotide polymorphisms, which decreased after selection; therefore, the π value was less in the selection than in the control line (Figure 4B). The results show that the two populations were strongly selected near the location of chr27:6,000,000, and rs312715211 was in this region. We have calculated the single point π value of rs312715211, as shown in Table 3. Taking all the results together, rs312715211, located on intron2 of *ZNF652*, reaches the thresholds of GWAS, F*_ST_*, and π; therefore, it can be considered a key variant associated with AFP.

### 3.4. rs312715211 Can Affect the Activity of ZNF652 Promoter

We speculated that rs312715211 affects the expression of *ZNF652*, so a double-luciferase validation was performed to verify the regulation of rs312715211 on expression of *ZNF652*. Then, PGL4.18/ZNF652pro/TT/AA dual-luciferase plasmids were transferred into 293T cells. When rs312715211 was wild genotype (TT), ZNF652 promoter activity was not changed (Figure 5A). rs312715211 significantly increased the activity of the *ZNF652* promoter, when the rs312715211 wild genotype TT was mutated to AA. The binding of transcription factors, meanwhile, was predicted using the PROMO online website after the rs312715211 mutation and wild-type. We found that most transcription factors were changed after mutation at this site (Figure 5B), which may affect gene expression. The results show that the rs312715211 region not only acts as an enhancer, but may be the transcription factor’s binding region.

### 3.5. rs312715211 Mutation Can Increase AFP and AFW and Decrease ECW

AFP consists of AFW and ECW. We compared the genotype frequency and phenotype (AFP, AFW, and ECW) of individuals carrying wild or mutant genotypes of rs312715211 SNP. According to these results, rs312715211 caused significant differences in the AFP, AFW, and ECW phenotypes between the selected and control lines. We also confirmed that rs312715211 can increase AFW and reduce ECW, thereby affecting AFP (Figure 6B). Both the genotype frequency and the phenotype associated with this SNP changed significantly after selection. The frequency of rs312715211 in the wild genotype increased after selection, as did AFP. The frequency of rs312715211 in the mutant genotype decreased after selection, while AFP increased (Figure 6A).

### 3.6. Identification of Candidate Genes and Pathways Related to AFP

To identify the candidate genes and pathways related to AFP, we widened the field of investigation. The top 1% of SNPs associated with AFP were screened accordingly, based on the GWAS results, and a total of 4736 genes were annotated and subjected to KEGG pathway enrichment analysis. KEGG enrichment analysis indicated that 52 pathways were significantly enriched (*p <* 0.05) (Figure 7 and Appendix A). Among them, the metabolic pathways, MAPK, neuroactive ligand-receptor interaction, and calcium signaling pathways were the most significant. Furthermore, some pathways related to fat or carbohydrate metabolism were significantly enriched, including PPAR, adipocytokine, glycerophospholipid metabolism, glycolysis/gluconeogenesis, and other glycan degradation signaling pathways. Unfortunately, *ZNF652* was not enriched in the related pathways. However, studies have shown that T*GFB1* and *EGFR* are the target genes of *ZNF652* and can be directly recruited and bound [25]. We also found that *EGFR* and *TGFB1* were the target genes of *ZNF652* and significantly enriched in 13 pathways (*p <* 0.05) (Table 4). Among these pathways, *EGFR* and *TGFB1* were simultaneously enriched in the MAPK and FoxO signaling pathways.

### 3.7. ZNF652 May Regulate Abdominal Fat and BW through MAPK/FoxO Signaling Pathways

To verify the above results, WGCNA was performed on the transcriptomes of the high and low AFP chickens. We found that *ZNF652* was expressed less in high AFP group than in the AFP low group, consistent with the dual-luciferase result (Figure 8A,B). We also found that AFP, AFW, BW, and ECW were simultaneously mapped in a significant module (darkseagreen2) (*p* < 0.05) (Figure 8C). Although *ZNF652* was not enriched in this significant module, it was co-expressed with classic lipid metabolism genes, such as *LIPN1, LIPN2, GATA3, PPARG, DGKQ,* and *DGKE*. Another KEGG analysis was performed on the enriched genes in the significant module, and a total of 19 significantly pathways were enriched (*p* < 0.05). Of these, the MAPK and FoxO signaling pathways were consistent with our above results (Figure 8D). In conclusion, we speculate that *ZNF652*, similar to the lipid metabolism genes, plays an important role in fatty acid degradation and may intervene in the MAPK and FoxO signaling pathways by binding with target genes, thus affecting body weight by decreasing AFP and AFW.

## 4. Discussion

Excessive abdominal fat in broilers not only reduces reproductive performance and causes metabolic diseases, but also reduces meat quality [26,27]. An increasing number of researchers are focusing on the genetic mechanisms underlying abdominal fat deposition. In this study, the heritability of AFP was 0.64, similar to the results reported by Demeure, Duclos et al. [7] and Chabault, Baéza et al. [6].

With the application of GWAS and selection signals, AFP in chickens has been fully utilized and SNPs and QTLs associated with abdominal fat have been identified [28]. According to the database, 292 QTLs and numerous SNPs were located in abdominal fat traits [2]. These QTLs and SNPs were verified in several broiler breeds and were significantly associated with fat deposition [29,30]. We attempted to identify the major genetic markers related to abdominal fat and growth traits in broilers. GWAS, F*_ST_*, and π analysis identified one significant SNPs on chromosome 27, and the SNP differed significantly between wild and mutant individuals in phenotype (*p* < 0.05). This is consistent with Zhang [3] and suggests that the SNPs are important genetic markers for reducing abdominal fat deposition in broilers.

As a result, rs312715211 conformed with our expectation. The SNP was mapped to ZNF652 and may simultaneously regulate abdominal fat deposition and growth development in chickens. rs12715211 can increase AFP, AFW, and BW in broilers after mutant. After selection, AFW significantly increased and BW significantly decreased. Moreover, rs312715211 significantly increased *ZNF652* promoter activity after mutant. In order to improve chicken breeding and increase its economic benefits, we should select wild homozygous individuals from rs312715211 for breeding and eliminate mutant homozygous individuals, so as to achieve increased body weight and reduce abdominal fat.

In the target region, *ZNF652* and *IGF2BP1* may be important for the control of abdominal fat deposition in broilers. *IGF2BP1* (insulin-like growth factor 2 mRNA-binding protein 1) can bind *IGF2* mRNA and is a member of the single-stranded RNA-binding protein family [31]. *IGF2BP1* can function by affecting cell proliferation, migration, and apoptosis [32]. *IGF2BP1* knockout mice can reduce Ramp3 mRNA expression; thus, *IGF2BP1* indirectly affects glucose and lipid metabolism [33]. In chicken pan-genomic studies, a high expression of *IGF2BP1* lead to high weight [34]. *IGF2BP1* can increase body weight and affects feed efficiency in ducks [35], promotes adipocyte proliferation and differentiation, and regulates expression of genes related to fatty acid metabolism in broilers [36]. *IGF2BP1* is also associated with tumors, cancer, dwarfism, and other diseases, but its mechanism of fat regulation requires further study.

*ZNF652* is the classical C2H2 zinc finger DNA binding protein [37] and an inhibitor of gene transcription and plays a primary role in tumor invasion [25]. *ZNF652* may be involved in human lipid and carbohydrate metabolism, e.g., sex hormone binding globulin (SHBG) [38] and may be related to body weight and bone growth [39,40]. However, its fat and growth regulatory mechanism in chickens is not currently known. It has been reported that the inhibition of *ZNF652* can increase the expression of *EGFR* and *TGFB1*, and *EGFR* can regulate fat deposition by regulating fatty acid synthase [38,41]. The inhibition of *EGFR* expression can significantly reduce BW and subcutaneous and abdominal fat mass in mice, and more importantly, *SREBP-1* and *FASN* expression decreased after *EGFR* inhibition [42]. *EGFR* is an epidermal growth factor receptor that regulates fat metabolism genes, especially *PPAR*, and the inhibition of *EGFR* can reduce the expression of adipose synthase [41]. *TGFB1* (transforming growth factor β) [43] can bond with the *SMAD* family. Our studies showed that AFW and BW decreased after *ZNF652* expression increased, consistent with the results after *EGFR* inhibition. Previous reports suggest the inhibition of the *TGFB1* gene by *PPARG* activation, and *PPARG* can inhibit cell proliferation through the TGF-β1/Smad2/3 signaling pathways [44]. Therefore, we hypothesized that the effect of *ZNF652* was associated with *EGFR, TGFB,* and *PPARG* by targeting the TGF-β1/SMad2/3 signaling pathways and MAPK/FoxO signaling pathways, resulting in reduced proliferation of adipocytes and accelerated fatty acid oxidation.

## 5. Conclusions

Our findings elucidated the genetic basis of chicken AFP. rs312715211 in the ZNF652 gene is the key variant associated with AFP of chickens, and the wild genotype is the favorable genotype to lower AFW and heighten BW. Additionally, ZNF652 is the key gene related to AFP by affecting BW and AFW. These data provide a new insight into our understanding of the genetic mechanisms underlying abdominal fat deposition in chickens and will aid in the breeding of broilers with lower AFP. In the future, the further study of these genetic variations may be applied in marker-assisted selection to reduce abdominal fat deposition in broilers.

## Figures and Tables

**Figure 1 biology-11-01849-f001:**
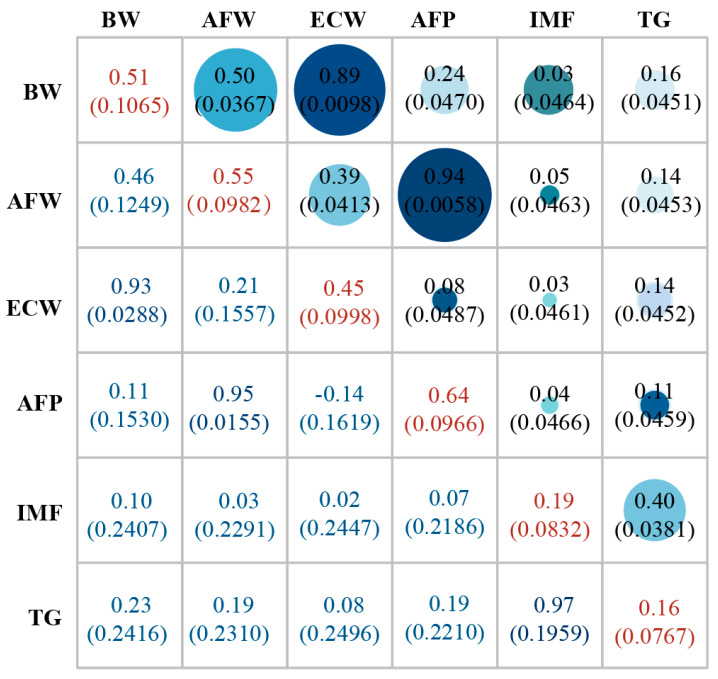
Genetic parameters and phenotypes. The upper triangle is phenotypic correlation coefficient (SD, standard error), the lower triangle is genetic correlation coefficient (SD, standard error), and the diagonal is heritability (SD, standard error).

**Figure 2 biology-11-01849-f002:**
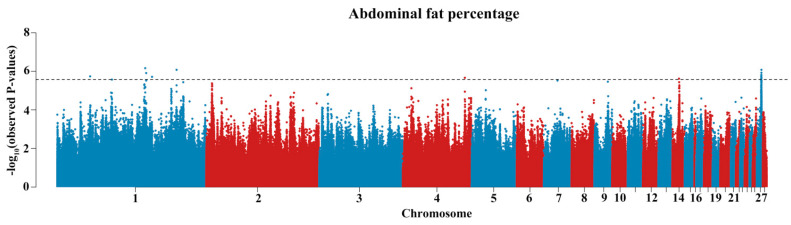
Manhattan plot of the genome-wide association study of abdominal fat percentage. The Manhattan plot indicates −log10 (observed *p*−values, *y*−axis) for genome-wide SNPs plotted against their respective locations on the genome. The horizontal red lines indicate the suggestive significant (2.64 × 10^−6^) thresholds.

**Figure 3 biology-11-01849-f003:**
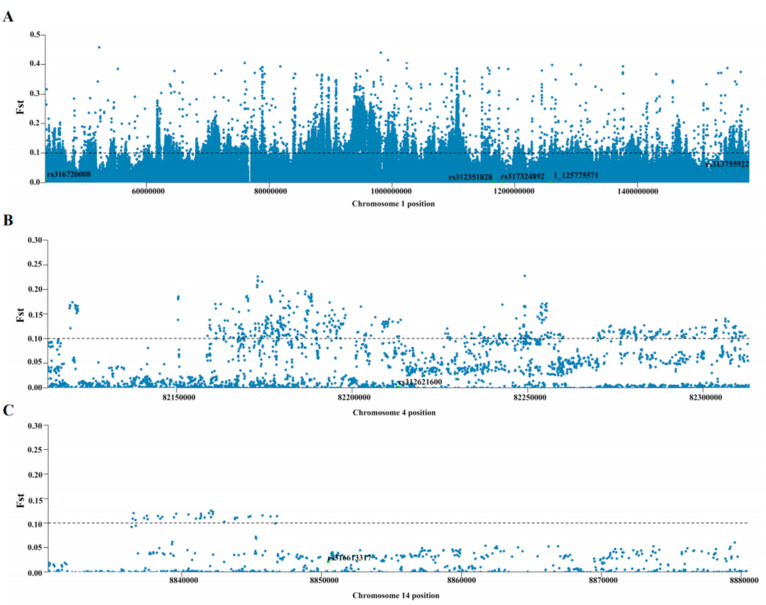
Locations of the candidate SNPs on chromosomes 1 and 14. (**A**) F*_ST_* of chromosome 1. (**B**) F*_ST_* of chromosome 4. (**C**) F*_ST_* of chromosome 14. Green represents significant SNPs located by genome-wide association study (GWAS), F*_ST_* threshold line: 0.1.

**Figure 4 biology-11-01849-f004:**
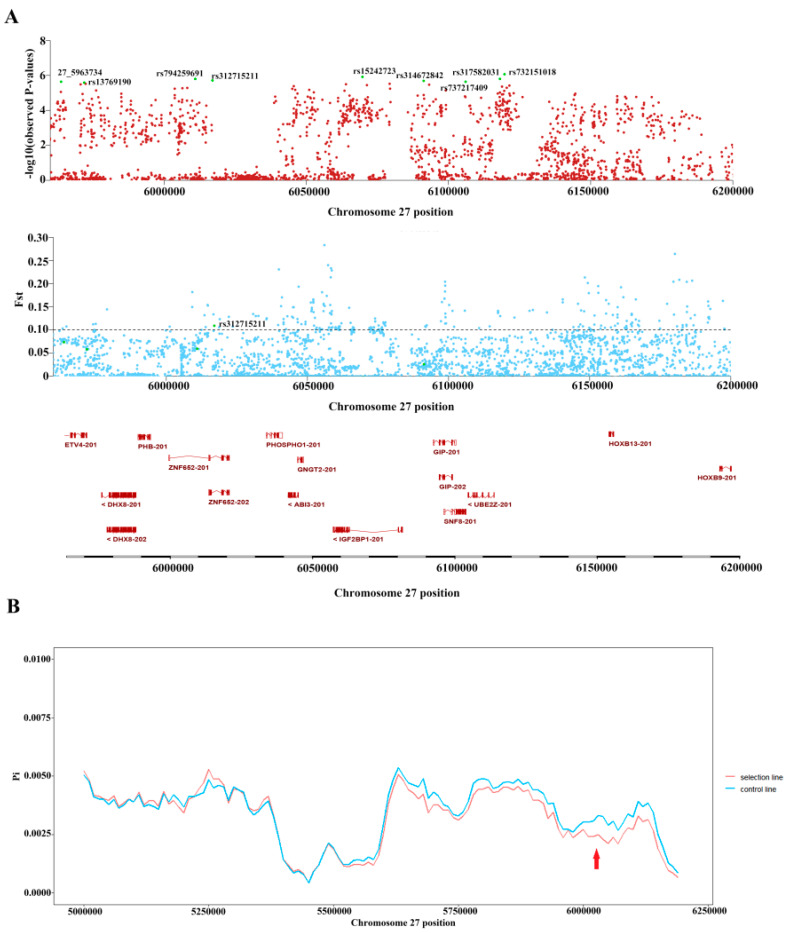
Locations of the candidate regions F*_ST_* and π. (**A**) Localized genome−wide association study (GWAS) and F*_ST_*. Green represents significant SNPs located by GWAS (Chr27: 5,963,734–6,119,680). Only rs312715211 reached the F*_ST_* threshold line. (**B**) Single nucleotide polymorphisms (π-values) in significant regions.

**Figure 5 biology-11-01849-f005:**
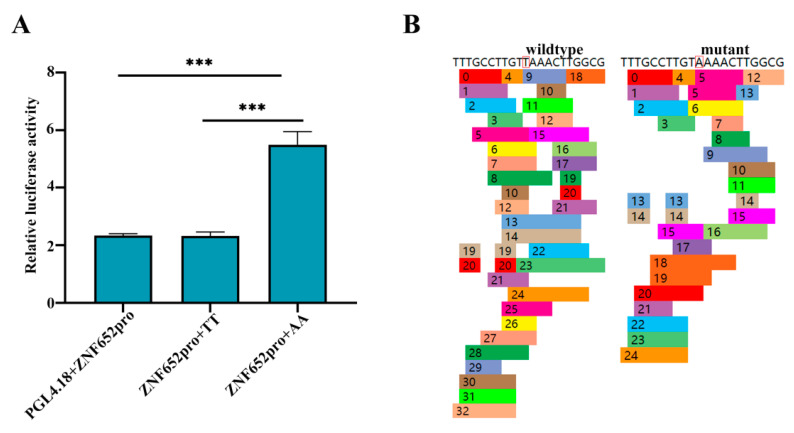
rs312715211 can regulate *ZNF652* promoter activity. (**A**) Fluorescence intensity of dual-luciferase, *ZNF652* promoter activity was significantly increased after site mutation, *** *p* < 0.001. (**B**) Prediction of wildtype and mutant transcription factors. All of the transcription factors were changed after rs312715211 mutated to AA.

**Figure 6 biology-11-01849-f006:**
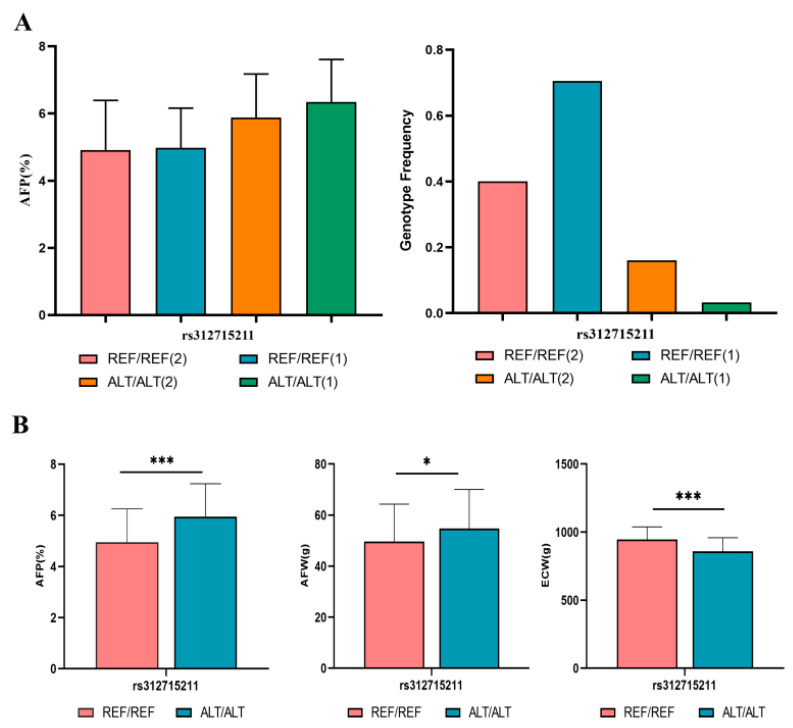
Genotype frequency and phenotype change of rs312715211 between the selected line and control line. (**A**) Phenotype and genotype frequency of candidate SNPs in the (1) selection line: 252, (2) control line: 268. (**B**) Candidate SNP phenotypes abdominal fat percentage (AFP), abdominal fat weight (AFW), and eviscerated carcass weight (ECW) in wild and mutant lines. AFP, AFW, and ECW were significantly increased after rs312715211 mutation, * *p* < 0.05, *** *p* < 0.001.

**Figure 7 biology-11-01849-f007:**
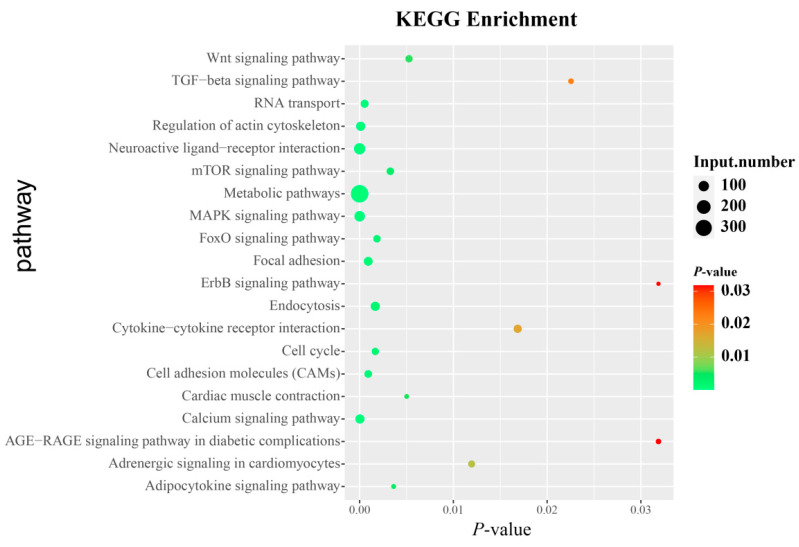
Kyoto Encyclopedia of Genes and Genomes (KEGG) analyses of annotated genes at the top 1% SNPs. The picture shows 20 KEGG pathways. Y represents the pathway, and X represents the rich factor. Size and color of the bubble represent the amounts of differentially expressed genes enriched in the pathway and the enrichment significance, respectively. The top 1% of SNPs were derived from the results of the genome-wide association study, and the analyzed individuals contained selection line: 252, control line: 268.

**Figure 8 biology-11-01849-f008:**
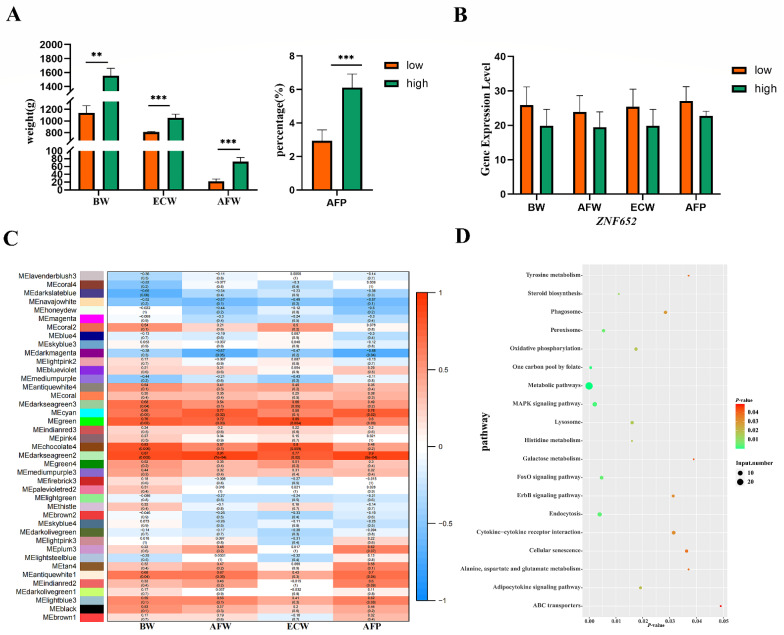
Expression levels of *ZNF625* in high and low abdominal fat percentage (AFP) groups. (**A**) Phenotypes of the high and low AFP groups, ** *p* < 0.01, *** *p* < 0.001. (**B**) *ZNF652* expression levels in the high and low AFP groups. (**C**) Heatmap showing module-trait associations. Each cell contains the corresponding correlation and *p*-value in parentheses. Red and blue colors represent positive and negative correlations. (**D**) Kyoto Encyclopedia of Genes and Genomes analyses of significantly module (darkseagreen2).

**Table 1 biology-11-01849-t001:** Descriptive AFP, AFW, ECW statistics at different lines.

Trait	Group	N	Mean (SD)	CV (%)	Min	Max	*p*-Value
IMF(g)	control line	252	1.84 ± 0.56	29.93	0.47	3.52	<0.0001
	selection line	268	2.15 ± 0.64	30.54	0.74	4.61	
TG(mg/g)	control line	252	3.51 ± 0.85	28.49	1.65	6.08	<0.0001
	selection line	268	3.92 ± 1.12	24.24	1.73	8.60	
AFP (%)	control line	252	5.21 ± 1.28	24.50	1.63	8.81	0.7453
	selection line	268	5.30 ± 1.26	23.79	2.09	8.37	
AFW (g)	control line	252	48.94 ± 14.26	29.14	8.60	92.70	<0.0001
	selection line	268	54.77 ± 14.96	27.31	18.20	96.30	
ECW (g)	control line	252	882.08 ± 86.67	9.83	684.2	1091	<0.0001
	selection line	268	973.66 ± 84.94	8.72	740.4	1206.4	

N, number of samples. SD, standard deviation. CV, coefficient of variance.

**Table 2 biology-11-01849-t002:** Summary of SNPs that reached the suggestive significance threshold on the genome.

SNP	CHR	Position	ALT/REF	MAF	β (SE) ^1^	*p*-Value	PVE	Distance ^2^	Gene
rs316720008	1	43,577,050	C/T	0.203	−0.6383165(0.1291873)	0.00000185	4.13%	intron2	*lncRNA*
rs312351828	1	116,670,617	T/C	0.126	−0.8267067(0.1729973)	0.0000007	5.35%	intron29	*DMD*
rs317324892	1	117,928,954	A/T	0.065	0.4439148(0.09234324)	0.00000124	4.89%	intron9	*IL1RAPL1*
1_125775571	1	125,775,571	G/C	0.117	0.5044212(0.1054399)	0.00000196	3.77%	D51254	*ARHGAP6*
rs313755922	1	158,194,105	C/G	0.434	0.5432101(0.1122472)	0.000000842	4.22%	intron1	*DACH1*
rs312621600	4	82,213,271	A/G	0.2	−0.5663142(0.1179772)	0.00000219	4.13%	U1957	*GRK4*
rs316613317	14	8,850,467	G/C	0.231	−0.5989047(0.1222303)	0.00000244	4.35%	intron7	*SYT17*
27_5963734	27	5,963,734	G/A	0.279	−0.5798911(0.1171215)	0.0000023	3.73%	intron2	*ETV4*
rs13769190	27	5,971,903	T/C	0.19	−0.5848534(0.1187516)	0.00000258	5.41%	D582	*ETV4*
rs794259691	27	6,010,935	C/T	0.191	−0.5642823(0.1175598)	0.00000157	5.58%	intron1	*ZNF652*
rs312715211	27	6,017,027	A/T	0.285	−0.5775431(0.1196338)	0.00000192	4.29%	intron2	*ZNF652*
rs15242723	27	6,069,759	A/G	0.28	−0.5986492(0.1190552)	0.00000121	4.49%	intron12	*IGF2BP1*
rs314672842	27	6,091,289	G/A	0.216	0.4740275(0.09803836)	0.00000203	4.06%	intron2	*GIP*
rs737217409	27	6,106,019	T/C	0.24	0.507996(0.1029385)	0.0000023	3.83%	intron2	*UBE2Z*
rs317582031	27	6,118,076	T/C	0.227	0.5165271(0.1051145)	0.00000157	3.34%	D4347	*UBE2Z*
rs732151018	27	6,119,680	A/G	0.241	0.5115049(0.1063718)	0.000000856	4.04%	D5951	*UBE2Z*

^1^ SE values are reported in parentheses. ^2^ U = upstream, D = downstream.

**Table 3 biology-11-01849-t003:** Candidate SNP and Gene.

SNP	CHR	Position	F*_ST_*	^1^ π (Group1)	^2^ π (Group2)	Gene
rs312715211	27	6,017,027	0.108971	0.272997	0.471467	*ZNF652*

^1^ group1, selection line, ^2^ group2, control line.

**Table 4 biology-11-01849-t004:** KEGG of pathway.

Pathway	Pathway Code	Corrected *p*-Value	Genes
MAPK signaling pathway	gga04010	0.000001450	*EGFR, TGFB1, PPP3CA, NGFR,* etc.
Calcium signaling pathway	gga04020	0.000031000	*EGFR, MCU, PPP3CC, STIM1,* etc.
Regulation of actin cytoskeleton	gga04810	0.000112941	*EGFR, CHRM5, MYLK2, MRAS,* etc.
Focal adhesion	gga04510	0.000916584	*EGFR, FYN, MYLK2, COL9A3,* etc.
Adherens junction	gga04520	0.001264711	*EGFR, BAIAP2, PTPRM, PTPRJ,* etc.
Endocytosis	gga04144	0.001669978	*EGFR, RAB7A, GRK5, ZFYVE9,* etc.
FoxO signaling pathway	gga04068	0.001849407	*TGFB1, EGFR, BCL6, CREBBP,* etc.
Cytokine-cytokine receptor interaction	gga04060	0.016863281	*TGFB1, IL5, EDAR, BMP7,* etc.
TGF-beta signaling pathway	gga04350	0.022547651	*TGFB1, SMAD9, SMAD5, DCN,* etc.
AGE-RAGE signaling pathway in diabetic complications	gga04933	0.031877403	*TGFB1, COL4A1, AKT3, NOX1,* etc.
ErbB signaling pathway	gga04012	0.031877403	*EGFR, GSK3B, PAK1, PAK3,* etc.
Gap junction	gga04540	0.037408823	*EGFR, DRD2, ADCY9, CDK1,* etc.

## Data Availability

The whole-genome resequencing datasets generated in this study were submitted to https://bigd.big.ac.cn/gsa (accessed on 10 July 2022), with IDs CRA002643 and CRA00265. The RNA-resequencing datasets of this study have been deposited at https://bigd.big.ac.cn/gsa/, accessed on 10 July 2022, (accession data code CRA006031).

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
