# Peer review of "Genome-Wide Association Study Revealed the Effect of rs312715211 in ZNF652 Gene on Abdominal Fat Percentage of Chickens"

_biology, 2022, doi:10.3390/biology11121849_

Round 1

Reviewer 1 Report

1. P 2 line 87 Blood samples were collected from the chicken's wings... 

It's better than Blood samples extracted from the chicken's wings...

2. please check for the Table that reports the P-value. Maybe write the number not used in the format of x.xxE-xx. 

thank you 

Author Response

Reviewer 1

1.P 2 line 87 Blood samples were collected from the chicken's wings...

It's better than Blood samples extracted from the chicken's wings...

Respond: Thank you for your suggestion, we was modified the “extracted” to “collected” in Line 91.

2.please check for the Table that reports the P-value. Maybe write the number not used in the format of x.xxE-xx.

Respond: Thank you for your suggestion, We've changed all the format of P value in Table1 and Table2.

Reviewer 2 Report

Comments to the Authors:

The manuscript is clearly presented and well conducted in its results, however, the title does not reflect the findings found through AFP analysis using a Genome-wide Association Study (GWAS) and fixation (FST) indices identified SNP rs312715211 and KEGGanalysis suggested that ZNF652 may reduce AFW and BW.  Therefore, the Title of the paper is more of an association according to the analyses performed and not so much of a claim that it affects abdominal fat percentage, as no experiment was performed to demonstrate this affectation.

Title section:

Change title according to the comment above.

Abstract section:

Describe the acronym KEGG

Reviewer 3 Report

My personal congratulation to the authors for the excellent study and the quality of the manuscript. I could not find any significant concern in this study, and the only suggestion is wording "SNP variants" in the title, otherwise "affects", rather than affect, still in the title.

Author Response

Reviewer 3

My personal congratulation to the authors for the excellent study and the quality of the manuscript. I could not find any significant concern in this study, and the only suggestion is wording "SNP variants" in the title, otherwise "affects", rather than affect, still in the title.

Respond: Thank you for your reminder. According to the suggestion from Reviewer 2, we changed the title “ZNF652 and its SNP variant affect the abdominal fat percentage of chickens” to “Genome-wide association study revealed the effect of rs312715211 in ZNF652 gene on abdominal fat percentage of chickens”, and this revision will be more appropriate to the content of the article.